

# Structure and diffusion of molten alkali carbonate salts at the liquid-vacuum interface

Gerrick E. Lindberg

Department of Applied Physics and Materials Science, Department of Chemistry and Biochemistry, Center for Materials Interfaces in Research and Applications, Northern Arizona University, Flagstaff, AZ, United States of America

## ABSTRACT

The liquid-vacuum interface of molten alkali carbonate salts is studied with molecular dynamics simulations. Three salts comprised of $Li_xNa_yK_zCO_3$ near their respective eutectic concentrations are considered to understand the distribution of ions relative to a liquid-vacuum interface and their diffusivity. These simulations show that each of the cations accumulate at the interface preferentially compared to carbonate. The cation ordering is found to inversely correspond to cation radius, with K being the most likely occupant at the surface, followed by Na, Li, and then the anion. Similar to other studies, the carbonate is found to diffuse more slowly than the cations, but we do observe small differences in diffusion between compositions that present opportunities to optimize ion transport. These results hold consequences for our understanding of ion behavior in molten carbonate salts and the performance of devices employ these electrolytes.

## INTRODUCTION

Molten carbonate salts have been studied in many contexts because of their occurrence in natural environments, in many engineered materials, and as components in various devices (*Gaune-Escard & Haarberg, 2014*). Alkali carbonate melts are specifically attractive for use in many applications because they have low vapor pressures, are easy to contain, and are generally environmentally safe (*Maru, 1984*; *Gaune-Escard & Haarberg, 2014*). A significant amount of interest has been motivated by electrochemical devices and chemical separation technologies, which can be designed to have high efficiency, resilience from fouling, and low material costs (*Maru, 1984*; *Dicks, 2004*; *Kirubakaran, Jain & Nema, 2009*; *Wade et al., 2011*; *Roest et al., 2017*). The performance of such devices have been studied in addition to the bulk behavior of individual components of relevant systems, but the thermodynamics and dynamics of alkali carbonate electrolytes near the relevant interfaces are not as well understood. The behavior of interfaces generally, and ions at liquid-vacuum interfaces, have received significant attention as regions featuring interesting manifestations of physical principles and as domains where distinct chemistries can occur (*Allara, 2005*; *Kumar, Knight & Voth, 2013*; *Soniat, Kumar & Rick, 2015*; *Tse et al., 2015*; *Bastos-González*

Corresponding author
Gerrick E. Lindberg,
gerrick.lindberg@nau.edu

*et al., 2016; Gutiérrez et al., 2018*). The purpose of this work is to apply similar methods to characterize the liquid-vacuum interface of three molten alkali carbonate electrolytes.

These electrolytes present many aspects that are of a general interest to chemists in many subdisciplines. Their performance and feasibility for use in molten carbonate fuel cells is of particular relevance because the operation of such devices relies on both the structure and transport of molecules at interfaces and the bulk within the device. In a typical molten carbonate fuel cell oxygen and carbon dioxide gases are fed to the cathode where $O_2$ is reduced and carbonate is formed. The carbonate ion is then transported through the electrolyte to the anode where carbonate reacts with hydrogen gas. The hydrogen is reduced, carbon dioxide is reformed, and water is produced. Therefore, the performance of these fuel cells depend on the efficient uptake of feed gases at an interface, transport of carbonate through the bulk electrolyte, and release of product gases at an interface. It is therefore important to characterize the bulk and interfacial behavior of these electrolytes.

Numerous computational studies have considered the bulk behavior of ion transport in these electrolytes (*Habasaki, 1990; Koishi et al., 2000; Costa, 2008; Vuilleumier et al., 2014; Ottochian et al., 2016; Corradini, Coudert & Vuilleumier, 2016*). Habasaki studied carbonate salts with Li or Na cations and found that the anion is significantly more mobile with the smaller cation, which they found is related to the ionic radii (*Habasaki, 1990*). Koishi and coworkers looked at carbonate salts with Li and K cations and found that carbonate diffusion is maximized when Li proportion is highest (*Koishi et al., 2000*). Costa and Ribeiro also found that carbonate diffuses fastest when there is more Li than K, but their trend is less clear, which they attribute to a small box size and higher system density (*Costa, 2008*). Corradini and coworkers generally found behavior of Li and K carbonate salts similar to other works, but interestingly postulated that ionic diffusion of cations and anions might be anticorrelated resulting in smaller ionic conductivities than would be anticipated from the diffusion constants and the Nernst-Einstein relationship (*Tissen, Janssen & Eerden, 1994; Corradini, Coudert & Vuilleumier, 2016*), which was also observed by *Vuilleumier et al. (2014)*. These studies, however, do not consider the thermodynamics or transport of alkali carbonate salts in the interfacial regions that are important for the transport of ions through electrochemical devices.

Interfacial behavior of these molten alkali carbonate salts has been considered in a few studies (*Roest et al., 2017; Gutiérrez et al., 2018; Gutiérrez et al., 2019*). Roest and coworkers studied the behavior of Li, Na, and K carbonate salts at charged and neutral interfaces (*Roest et al., 2017*). They have identified distinct mass and charge profiles in the molten liquid near the interface and the ions are found to diffuse more slowly in close vicinity of the interface. *Gutiérrez et al. (2018)* and *Gutiérrez et al. (2019)* have studied molten $LiCO_3$ and a eutectic mixture of $LiNaKCO_3$ at an interface with carbon solids, gases, and vacuum. They find that the ions do arrange at the interface as defined by the Gibbs dividing surface. The Li and carbonate ions are found to have similar profiles outside of the Gibbs dividing surface (in the vacuum region), but Li and carbonate are found to be weakly layered below the interface in the liquid. In molten $LiNaKCO_3$, they find a more complicated distribution of ions. The K is found to be most prevalent in the region outside the Gibbs dividing

**Table 1  Cation fractions for the eutectic salts used in this work.**

| System | Li | Na | K |
|---|---|---|---|
| LiNaKCO$_3$ | 0.435 | 0.315 | 0.25 |
| LiNaCO$_3$ | 0.53 | 0.47 | 0 |
| LiKCO$_3$ | 0.427 | 0 | 0.573 |

surface, and Li and Na are found just inside the Gibbs dividing surface. These studies leave questions about how salt composition affect ion and interface structure and dynamics.

It is with this in mind that this study examines the behavior of carbonate ions at a liquid-vacuum interface. The liquid-vacuum interface has been selected because sorption and desorption processes of feed and waste gases is poorly understood at a molecular scale, but crucial to the operation of devices using these materials. In this work, the structure and transport of molten Li, Na, and K containing carbonate salts are examined at the liquid-vacuum interface. We carefully examine local density profiles with respect to the interface using two definitions of the divide between the liquid and vacuum. These two definitions provide complementary perspectives on ion behavior at the boundary between the two phases, which will likely be useful for future studies. We also estimate the slab width, surface tension, and self-diffusion constant, and relate the observed values to the system composition.

## METHODS

Simulations were performed of three eutectic systems with chemical formulas of the form Li$_x$Na$_y$K$_z$CO$_3$, where $x + y + z = 2$. Redox processes are not considered in this work, so Li, Na, and K always refer to the cations and C and O to the constituents of the carbonate anion, which comprise the electrolyte. The fractions of each cation in each system considered in this work are listed in Table 1. Cation fractions instead of mole fraction or concentration are used to emphasize the relative cation amounts in each system. The three systems considered have unique elemental compositions (LiNaCO$_3$, LiKCO$_3$, or LiNaKCO$_3$), so the alkali metal subscripts are dropped to simplify identification. These systems have been selected because they are near the eutectic compositions (*Janz, 1967*), which permits for the salt to be molten and for device operation at the lowest temperatures.

Initial configurations were prepared by placing 1000 cations (according to the fractions in Table 1) and 500 carbonate anions randomly on a grid in a simulation cell with dimensions of 40 Å × 40 Å × 100 Å. Ions were placed on evenly spaced, 4 Å  grid points by choosing a species randomly within the constraints of the particular composition of interest. Cations were placed on the grid points and the carbon from carbonate was placed at the grid points with the oxygen atoms placed around it. The ions were positioned just below the liquid density so they form a so-called *slab* of liquid surrounded by a large vacuum region. The system is constructed so that the average liquid-vacuum interface is perpendicular to the $z$-axis (Fig. 1). This study uses the interaction parameters developed by Tissen and Janssen (*Tissen & Janssen, 1990*; *Janssen & Tissen, 1990*), sometimes called the JT model, which have been employed for many similar studies of molten alkali
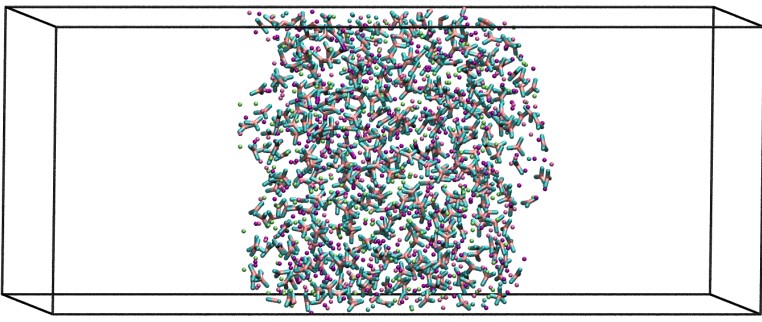

**Figure 1** **Snapshot of the LiNaKCO$_3$ simulation cell.** An example the slab and interfaces studied in this work.

carbonate salts and as such provides a robust body of literature for validation and reference (*Tissen & Janssen, 1990*; *Janssen & Tissen, 1990*; *Tissen, Janssen & Eerden, 1994*; (*Koishi et al., 2000*; *Ottochian et al., 2016*; *Wilding et al., 2016*; *Roest et al., 2017*; *Du et al., 2017*; *Du et al., 2019*; *Ding et al., 2018*; *Gutiérrez et al., 2018*). The JT model employs coulombic interactions with long-ranged interactions described by Ewald summation and Born-type repulsion parameterized from quantum mechanical calculations (*Tissen & Janssen, 1990*). Simulations were performed with the LAMMPS molecular dynamics package (*Plimpton, 1995*). Electrostatic interactions beyond 11.4 Å were calculated with the particle–particle particle-mesh with an accuracy of $10^{-5}$. The temperature was held constant using the Nosé-Hoover algorithm with a damping parameter of 500 fs and carbonate was kept rigid using the SHAKE algorithm as implemented in LAMMPS. Each system was heated from 0 to 1,200 K in 50 ps and then equilibrated at 1,200 K for 50 ps with a time step of 0.05 fs in the constant number of particles, temperature, and volume (NVT) ensemble. They were then further equilibrated in the NVT ensemble for 20 ns with a timestep of 0.5 fs. Therefore, the thermodynamic data presented was extracted from 10 ns NVT production simulations and the diffusion constants from 10 ns constant number of particles, volume, and energy (NVE) simulations. In these production simulations, configurations were saved every 0.5 ps and analysis included all frames from the corresponding NVT or NVE trajectory.

Molecular visualization was performed with Visual Molecular Dynamics (VMD) (*Humphrey, Dalke & Schulten, 1996*). Density profiles were calculated with in house scripts. Densities are normalized by dividing by the average density in the center of the respective slab to facilitate comparison between systems with different compositions. The Gibbs dividing surfaces are determined from the z-dimension density profile to find the two planes (one on each side of the slab) where the density is half the average bulk liquid density (*Gochenour, Heyert & Lindberg, 2018*). The interface, however, need not be viewed as a static plane, but can also be viewed as a dynamic, three-dimensional region specific to the underlying molecular configuration. This is similar to the distinction between 'sea level' and waves on the sea, where sea level is determined by averaging over local fluctuations to obtain a useful, but dramatically simplified, description. Analogously, it is useful to consider an interpretation of the interface that captures the local molecular-scale undulations. In this

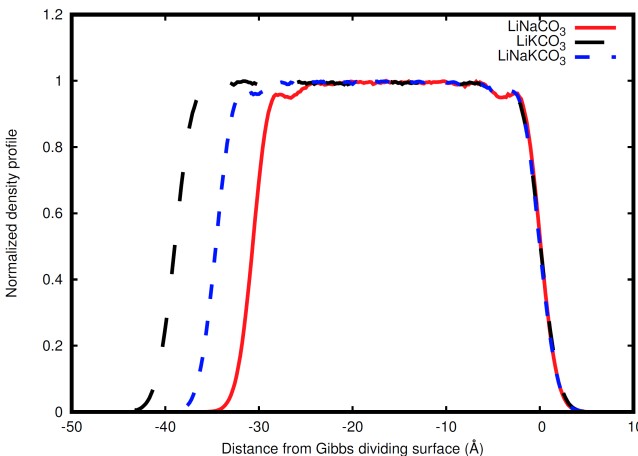

**Figure 2** **Normalized total density profile in the dimension perpendicular to the interface.** The densities are normalized by dividing by the average liquid density. For each system, the right Gibbs dividing surface is positioned at 0 Å and the left interface provides a sense of the slab width because the xy area of the slab remains constant. Expansion of the slab is highly correlated with increasing concentration of the larger radius K ion. Each system shows structure near each interface that corresponds to structure within the liquid.

work, we utilize the instantaneous interface scheme developed by Willard and Chandler (*Willard & Chandler, 2010*). Briefly, this method involves creating a coarse-grained density field and identifying points in this coarse grained density field at a density halfway between that of the two phases. These points can then be connected to identify the interface that separates the two phases. The long time average of the instantaneous interface is analogous to the Gibbs dividing surface. The instantaneous interfaces were calculated with an in-house script using a coarse graining length, $\xi$, of 1.5 Å. Self-diffusion constants are determined from the mean-squared displacement as calculated by CPPTRAJ with the Einstein relation (*Roe & Cheatham, 2013*).

## RESULTS

A snapshot of the equilibrated simulation setup for the $LiKCO_3$ system is shown in Fig. 1. The distribution of ions within each system are examined with density profiles perpendicular to the interface in Fig. 2. The organization of each element is isolated with atomic density profiles for each system in Fig. 3. The contribution of each element to the whole density profile is examined in Fig. 4. An example instantaneous interface for the $LiKCO_3$ system is shown in Fig. 5 and the distribution of instantaneous interface sites relative to the Gibbs dividing surface for each system are shown in Fig. 6. Finally, a histogram of the nearest distance between each element and the closest point on the instantaneous interface is shown in Fig. 7.
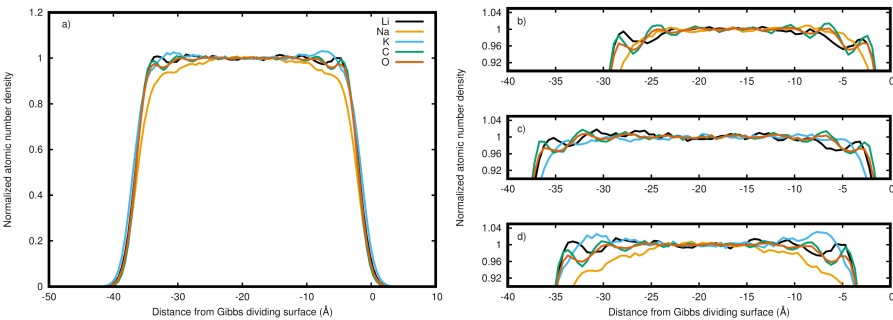

**Figure 3** **Normalized densities with respect to the right Gibbs dividing surface.** Li (black), Na (orange), K (blue), C (green), and O (red) in the (A) entire LiNaKCO$_3$ system and detailed views of the liquid region for (B) LiNaCO$_3$, (C) LiKCO$_3$, and (D) LiNaKCO$_3$ are shown. For each system, the right Gibbs dividing surface is positioned at 0 Å. The peaks and valleys are indicative of the atomic structuring near the surface.

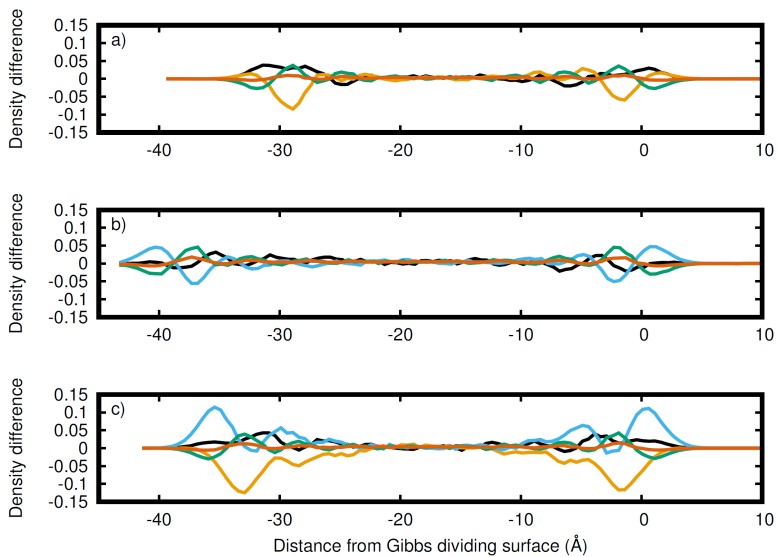

**Figure 4** **Atomic density differences from the total density profile in the dimension perpendicular to the Gibbs dividing surface.** Li (black), Na (orange), K (blue), C (green), and O (red) for the (A) LiNaCO$_3$, (B) LiKCO$_3$, and (C) LiNaKCO$_3$ systems are shown. These are the profiles in Fig. 2 with the whole system profiles in Fig. 1 subtracted. These plots emphasize the contributions of each element to the total density profile. For each system, the right Gibbs dividing surface is positioned at 0 Å.

## DISCUSSION

### Density profiles reveal interface-induced structure

Ion behavior in the vicinity of the liquid-vacuum interface was first examined with density distributions perpendicular to the plane of the interface. Figure 2 shows the normalized density along the coordinate perpendicular to the interface for each system considered. The most obvious difference observed in the simulations is an increase in slab width. This change in width is directly related to the amount of K, the largest cation considered, in the

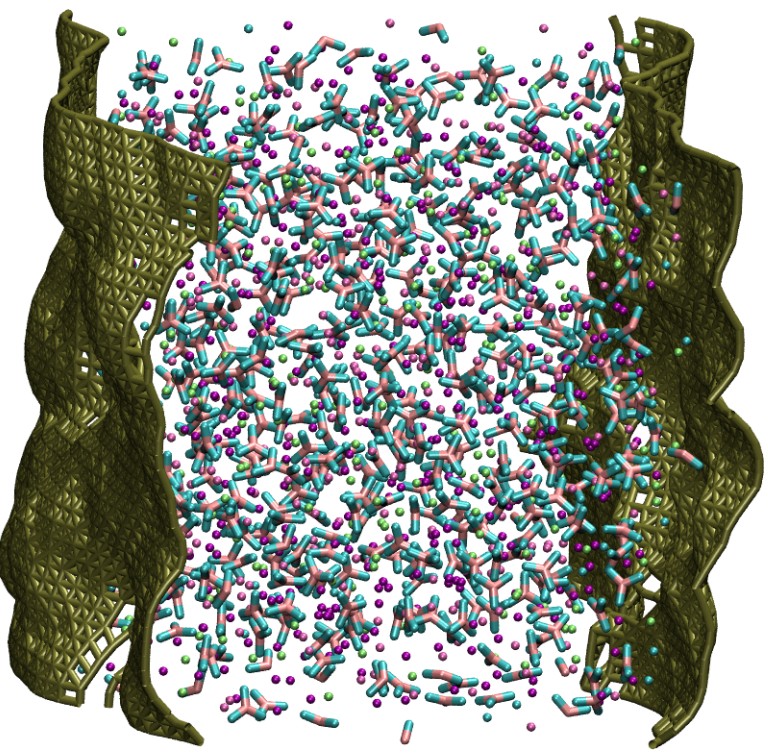

**Figure 5** **A representative snapshot of the LiKCO₃ system with the corresponding instantaneous interface.** The atoms comprising the salt are depicted in green, purple, blue, and pink and the instantaneous interface is shown in gold.

solution (Fig. 2). Near the interface there is structure in the densities. The liquid region is flat indicative of homogeneous liquid structure, the intermediate region between the liquid and vacuum shows two peaks, and finally there is a drastic decrease in density when moving into the vacuum. The peaks are most well-defined in the Na-containing systems (red and blue curves in Fig. 2), but these features are present in each system.

## Slab width and surface tension are highly correlated with system composition

The width of the slab can be defined as the distance between the Gibbs dividing surface on each side of the slab (Table 2). The slab width is found to be highly correlated with the size of the cations. Using ionic radii of 0.68, 0.97, and 1.33 Å for Li, Na, and K cations (*Weast, 1968*), an average cation radius can be calculated for each system

$$\langle r_{ion}\rangle = \chi^{Li} r_{ion}^{Li} + \chi^{Na} r_{ion}^{Na} + \chi^{K} r_{ion}^{K} \tag{1}$$

where $\langle r_{ion}\rangle$ is the average cation radius, $\chi^{X}$ is the fraction of the cations that are $X$, and $r_{ion}^{X}$ is the ionic radius of $X$. The slab width is found to be highly correlated with the average cation radius, with a linear coefficient of correlation ($R^2$) of 0.999. The individual cation sizes, however, are less correlated with slab width. The width is most weakly correlated with the cation fraction of Li, but significantly stronger correlation is observed with Na

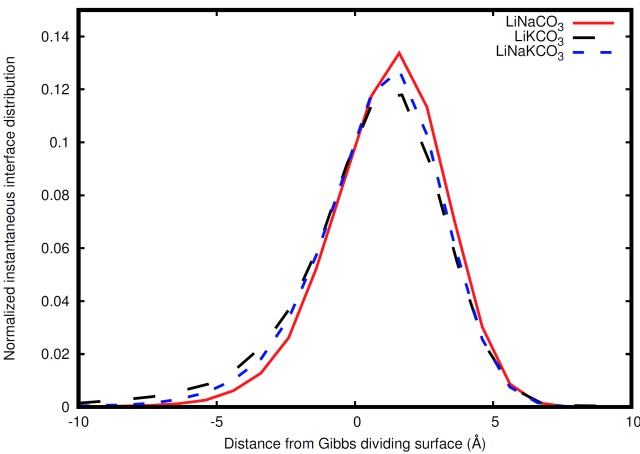

**Figure 6** **Distribution of instantaneous interface sites relative to the Gibbs dividing surface.** LiNaCO$_3$ (red), LiKCO$_3$ (black), and LiNaKCO$_3$ (blue) are shown.

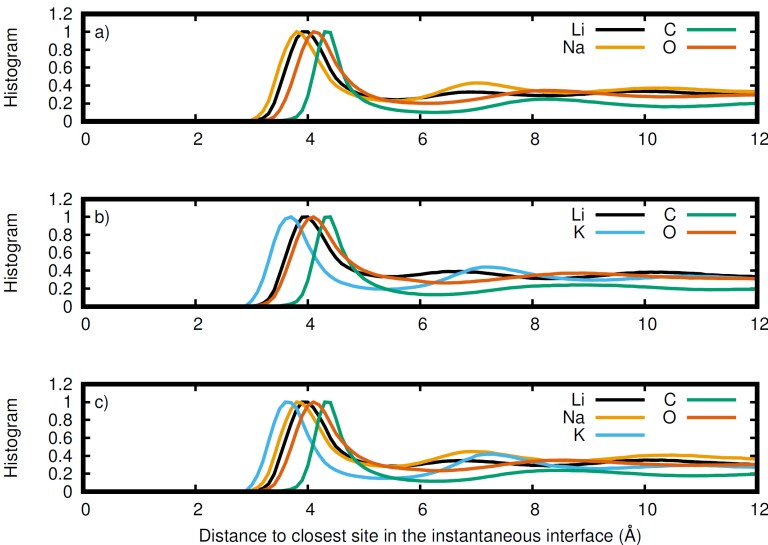

**Figure 7** **Normalized histogram of the distance of each element from the closest point on the instantaneous interface.** Li (black), Na (orange), K (blue), C (green), and O (red) for the (A) LiNaCO$_3$, (B) LiKCO$_3$, and (C) LiNaKCO$_3$ systems are shown.

and K cation fractions. The correlation coefficients are 0.780, 0.975, and 0.998 for Li, Na, and K, respectively. These conclusions are similar to the observation by Habasaki that anion-cation spacing is correlated with the cationic radii of Li and Na in carbonate salts (*Habasaki, 1990*), which indicates that molecular-scale spatial correlations can provide a good indication of larger bulk densities.

| Table 2 | Width and surface tension for each of the systems studied. | |
|---|---|---|
| System | Width (Å) | Surface tension (mN/m) |
| LiNaCO$_3$ | 30.7 | 137 ± 9 |
| LiKCO$_3$ | 39.0 | 125 ± 9 |
| LiNaKCO$_3$ | 34.6 | 130 ± 10 |

The surface tension can be estimated from the cell width and the diagonal components of the pressure tensor according to the expression

$$\gamma = \frac{L_z}{2} \left\langle P_{zz} - \frac{1}{2} \left( P_{xx} + P_{yy} \right) \right\rangle \tag{2}$$

where $\gamma$ is the surface tension, $L_z$ is the simulation cell width in the $z$ dimension, $P_{zz}$ is the component of the pressure tensor normal to the interface, and $P_{xx}$ and $P_{yy}$ are the tangential components of the pressure tensor (*Kirkwood & Buff, 1949*). This is similar to the surface tension protocol used by others (*Bhatt, Newman & Radke, 2004*; *Desmaele et al., 2019*). Surface tension values are shown in Table 2 and have not been reported previously for the JT model. Similar to the slab width discussed previously, the surface tension is found to be highly correlated with K concentration, slightly less with Na, and significantly less with Li. The surface tensions measured here seem to be systematically less than reported in the experimental literature (*Ward & Janz, 1965*; *Kojima et al., 2008*). For example, Kojima et al. report the surface tension of LiNaKCO$_3$ at 1200 K to be approximately 200 mN/m while the JT model yields a surface tension of 130 ± 10 mN/m (*Kojima et al., 2008*). Therefore, the JT model yields surface tension values that are about 35% smaller than those reported in the experimental literature (*Ward & Janz, 1965*; *Kojima et al., 2008*). The JT model appears to underestimate cohesive interactions, which is similar to previous studies that have shown a similar disagreement when the model density is compared to experiment (*Ottochian et al., 2016*; *Roest et al., 2017*; *Desmaele et al., 2019*). Some works have compensated for this discrepancy by performing their simulations at elevated pressure (*Ottochian et al., 2016*; *Roest et al., 2017*), but this is not possible here because the liquid is in contact with vacuum and therefore free to expand. Nevertheless, the literature provides significant evidence that this model provides a generally faithful description of molten alkali carbonates and as such lends confidence that the trends reported here are reliable. Additionally, the JT model has received such widespread usage that characterization of the surface tension provides important perspective on strengths and weaknesses of the model.

## Elemental structure at the Gibbs dividing surface

The normalized density in Fig. 2 includes all species in the system. It is interesting to decompose the density into the contributions from each elements to see if the ions accumulate differently at the interface. Therefore the density profile of each element was calculated to resolve these distributions near the interface. For example, the complete density distribution of each element in the LiNaKCO$_3$ system is shown in Fig. 3A. Similar to Fig. 2, the densities in Fig. 3 are normalized by dividing by the average density in the

middle of the slab. Figure 3A reveals the anticipated general trend of high density within the slab and no density outside it in the vacuum and the peaks and valleys indicate interfacial ordering of the individual ions in the elemental density profiles. This is similar to the entire system density profiles shown in Fig. 2. These features are difficult to resolve because of the large density difference between the liquid and vacuum, so Figs. 3B–3D, and d show detailed views of the liquid region of the elemental density profiles for each system. Each element is observed to have maxima and minima induced by proximity to the interface that are much larger than the subtle, apparently random wiggles in the middle of the slab. The similarity of the right and left interfaces indicates that the profiles are converged. Most notably, Na is observed (orange lines) to be depleted compared to the other elements near the Gibbs dividing surface (Figs. 3B and 3D). In the electrolyte without Na (Fig. 3C), K is also found to be depleted near the Gibbs dividing surface. The other K-containing solution (Fig. 3D) does not show similar depletion, but instead K has a significant maximum about 7 Å on the liquid side of the Gibbs dividing surface. Lithium is generally observed to be the cation with a maximum closest to the interface, which is generally similar in shape and location to the C peak of the anion. The C and O peaks are generally similar, with the O showing slightly less structuring. While some information about ordering of each element relative to the interface is observed, these features are nevertheless difficult to resolve. Therefore, it would be useful to further examine the local enrichment or depletion of each species relative to the interface for a clearer picture of ion distributions.

While Fig. 3 shows the structure of each atom at the Gibbs dividing surface in the liquid region, it is difficult to identify the atomic contributions to the total density profile and more specifically to see the atomic ordering within the interfacial region. Therefore, Fig. 4 depicts the difference between Figs. 3B–3D and the corresponding total density profiles in Fig. 2. Figure 4 shows that K has significant density at the surface and is the most prominent, when it is present. Conversely, as was observed in the analysis of Fig. 3, Na has a major depletion near the interface compared to its bulk density. The behavior of Li is less dramatic. When K is present, Li has a peak near the same distance from the interface. When K is not in the solution (Fig. 4A), then Li is the predominant cation at the Gibbs dividing surface. In all cases, C is depleted at the surface compared to the cations, but has a maximum just inside the surface. The error in these density profiles was estimated by breaking up the simulations into ten segments of equal length and calculating the error in the mean from these segments. Figure S1 depicts the normalized atomic density profiles with error for each element in the LiNaKCO$_3$ system. This shows that the error is smaller than peaks and valleys discussed, which indicates that the observed features are physically meaningful. Additionally, the error would be estimated to be even smaller than depicted because of agreement between the left and right sides of the interface, which are independent from each other. These findings are similar to those reported by Gutiérrez et al. who also showed ionic structuring near the Gibbs dividing surface (*Gutiérrez et al., 2018*). Next, these ionic arrangements will be examined further using a local definition of the interface.
## Instantaneous interface analysis reveals local fluctuations of the liquid surface

The interfacial analyses so far have been performed with respect to the Gibbs dividing surface. The Gibbs dividing surface is the plane that on average separates two distinct phases (*Gochenour, Heyert & Lindberg, 2018*), but the instantaneous interface method can provide a description of the interface with molecular features of the surface. An example of such an instantaneous interface is shown for the LiKCO$_3$ system in Fig. 5. In this work, we use the instantaneous interface to characterize fluctuations of the surface and elemental distributions in the vicinity of the instantaneous interface.

## Elemental structure at the instantaneous interface shows cations preferentially populate the surface

The instantaneous interface corresponds to the boundary between the molten salt liquid and vacuum for a particular molecular configuration. Therefore, analysis of the instantaneous interface and the underlying atomic distribution provides details about the behavior of the system in light of a specific arrangement of the atoms, rather than the time-averaged, global behavior described by the Gibbs dividing surface. Figure 6 shows a histogram of the instantaneous interface with respect to the Gibbs dividing surface. The instantaneous interface is found to not be symmetric or symmetrically distributed around the Gibbs dividing surface. Instead the asymmetric distribution has a maximum beyond the Gibbs dividing surface with a long tail extending into the liquid regime. Depending on the system, the instantaneous interface is found to extend between 5 to 10 Å into the liquid region. This is in agreement with the structure observed in Fig. 3, which shows structure in the atomic density distributions to approximately 10 Å into the liquid. There are differences between the solutions considered, with the LiNaCO$_3$ solution having the least pronounced skewness extending into the liquid and LiKCO$_3$ the most, which provides an estimate of the size of the interfacial region. It would be expected that larger solution surface tensions would correspond to a greater energy associated with deformation from a minimum surface area, but it appears that the subtle differences in surface tension are not large enough to have a clear effect on the distribution of instantaneous interface sites. Figure 7 shows a histogram of the shortest distance between each element and the instantaneous interface. The curves are normalized so they all have maximum values of 1 to simplify comparisons between species, despite different concentrations. The density profiles relative to the instantaneous interface in Fig. 7 provide a complementary perspective on ion structure near the interface to those observed with respect to the Gibbs dividing surface in Figs. 3 and 4. In Fig. 7, the cations are always more likely to be closer to the instantaneous interface than the anions. The ordering is distinct, with K being closest, then Na, Li, O, and finally C. It is interesting to note that the cation trend inversely follows the ionic radii. This indicates that the cations preferentially populate the surface of the instantaneous interface. Interestingly, these distributions are apparently independent of the electrolyte composition, and therefore possibly indicative of broader trends in ion behavior in these molten alkali carbonate electrolytes. The uniformity of the elemental distributions in each panel of Fig. 7 is distinct from the behavior observed with respect to the Gibbs dividing surface shown in Figs. 3 and

**Table 3** Self-diffusion constants of each ion in each of the systems studied.

| System | Li (Å²/ps) | Na (Å²/ps) | K (Å²/ps) | C (Å²/ps) |
|---|---|---|---|---|
| $LiNaCO_3$ | $4.0 \pm 0.3$ | $4.4 \pm 0.4$ | – | $1.5 \pm 0.1$ |
| $LiKCO_3$ | $3.3 \pm 0.3$ | – | $4.5 \pm 0.3$ | $1.8 \pm 0.2$ |
| $LiNaKCO_3$ | $3.2 \pm 0.3$ | $3.8 \pm 0.3$ | $4.2 \pm 0.4$ | $1.5 \pm 0.1$ |

4. This indicates that the dynamics of the interface can be affected by the composition, but the actual arrangement of the atoms is less sensitive.

## Anions diffuse slower than the cations

Finally, the transport of the ions is evaluated using the self-diffusion constants. Table 3 shows the diffusion constant of each species in each system. The diffusion constants are in general agreement with previous, similar studies (*Habasaki, 1990*; *Koishi et al., 2000*; *Costa, 2008*; *Vuilleumier et al., 2014*; *Ottochian et al., 2016*; *Corradini, Coudert & Vuilleumier, 2016*; *Roest et al., 2017*), however exact comparison with bulk diffusion constants is difficult since the systems described here are heterogeneous. Nevertheless, this work shows that carbonate diffuses slower than all of the cations. The cation self-diffusion is found to be proportional to cation size, with the larger ions diffusing faster. This is attributed to each cation having the same charge, so larger radii ions experience correspondingly weaker electrostatic interactions.

## CONCLUSIONS

Molecular dynamics simulations have revealed thermodynamic and dynamic properties of molten alkali carbonates near a liquid-vacuum interface. This work has shown that in three alkali carbonate salts, the alkali cations preferentially accumulate at the interface more than the anion. Additionally, the anions are found to diffuse much more slowly than the cations. Nevertheless, subtle differences are seen between the three solutions considered, which may be helpful in the selection of electrolyte compositions that yield the best performance in an electrochemical device. Intriguingly, the $LiKCO_3$ solution is found to yield the fastest carbonate diffusion while also permitting the closest approach of the anion to the interface. These differences could have significant effects on the performance of devices employing such electrolytes and therefore warrant study to understand and confirm these results.

## ACKNOWLEDGEMENTS

This work employed Northern Arizona University's Monsoon computing cluster, which is funded by Arizona's Technology and Research Initiative Fund.

### Funding

The authors received no funding for this work.

## Competing Interests

The authors declare there are no competing interests.

## Author Contributions

- Gerrick E. Lindberg conceived and designed the experiments, performed the experiments, analyzed the data, contributed reagents/materials/analysis tools, prepared figures and/or tables, performed the computation work, authored or reviewed drafts of the paper, approved the final draft.

## Data Availability

All information needed to fully reproduce the data in the article is available from the Methods section, where the simulation parameters, simulation protocols, and the analysis are described. The simulation software, LAMMPS, is open access and available for download from the developers.

## Supplemental Information

Supplemental information for this article can be found online at http://dx.doi.org/10.7717/peerj-pchem.3#supplemental-information.

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
