# Peer review of "Structure and diffusion of molten alkali carbonate salts at the liquid-vacuum interface"

_PeerJ Physical Chemistry, doi:10.7717/peerj-pchem.3_

## Round 0.1 · original submission · Major Revisions

Dear Dr. Lindberg,

Thank you for submitting your manuscript to PeerJ Physical Chemistry.
While I appreciate the impact and novelty your work may bring to the scientific community, I strongly recommend you to revise the manuscript according to the reviewers' suggestion and a point to point response in your revision is highly appreciated.

Once again, thank you very much for your submission.

Jiang Cui

Reviewer 1 ·

Basic reporting

'no comment'

Experimental design

'no comment'

Validity of the findings

'no comment'

Additional comments

1.The methods of the manuscript needs more detail. I suggest that you improve the description at lines 114- 120 to provide more justification for your study (specifically, you should give the
specific expression of potential function and detailed numerical value of the interaction parameters ).
2.In the manuscript, the simulation results lack a detailed calculation process. It is only based on similar methods in the literature, but the specific calculation method is not given. This makes it difficult for future readers to understand.
3.The description at lines110-112"Initial configurations were prepared by placing 1000 cations (according to the fractions in Table 1) and 500 carbonate anions randomly on a grid in a simulation cell with dimensions of 40 Å*40 Å*100 Å." Please give the answers to the following questions in the appropriate place in the manuscript.
a. What kind of random algorithm is used for random placement?
b. Are there multiple simulations and statistical analysis of the simulation results?
c. Are the results of multiple simulations consistent?
d. Whether the micro-simulation results can be extended to the macro-system?

Annotated reviews are not available for download in order to protect the identity of reviewers who chose to remain anonymous.

Reviewer 2 ·

Basic reporting

The English require substantial improvements. For example, line 36: are specifically; Line 81: they have identified; line 105: The considered three systems; line 112: were; line 113:was ; line 129: Visual Molecular Dynamics (VMD) ……
In the introduction, paragraph 3 required references to validate the statement.

Line 229 to 245: description of identifying the instantaneous interface should go to method part.

Experimental design

The purpose of this work is not very clear. Why are authors focused on liquid-vacuum interface not other interface? The potential application of the findings should be described more specifically.

How the total number of ions in the systems are determined? What is the initial liquid density?

The description of the system set up is not very clear. It is better to have a representative figure to show the system set up.

Validity of the findings

The simulation time is very short, and the equilibration of systems should be validated to ensure the sufficient sampling such as analysis of the density change with time

The observed peaks and drops are very small, and the fluctuation could be dramatic along the simulation. The density profile evolved with time should be analysis and error bar should be added to time-averaged density profile.

It is not clear how the results are analyzed. Is it analyzed using the last frame in the simulation or averaged over all the frames?

---

## Round 0.2 · accepted · Accept

The reviewers comments indicated that the revised manuscript should be accepted and published in PeerJ Physical Chemistry. Therefore, I'm pleased to accept your revised manuscript for publication.
Thank you.

Reviewer 1 ·

Basic reporting

'no comment'

Experimental design

'no comment'

Validity of the findings

'no comment'

Additional comments

The author has revised the manuscript as requested by the reviewer.The reviewer suggested accepting the manuscript.

Reviewer 2 ·

Basic reporting

no comment

Experimental design

no comment

Validity of the findings

no comment

Additional comments

My comments have been addressed and I recommend for publication.